# Diagnostic Performance of the Measurement of Skinfold Thickness for Abdominal and Overall Obesity in the Peruvian Population: A 5-Year Cohort Analysis

**DOI:** 10.3390/ijerph20237089

**Published:** 2023-11-21

**Authors:** Cristian Rios-Escalante, Silvia Albán-Fernández, Rubén Espinoza-Rojas, Lorena Saavedra-Garcia, Noël C. Barengo, Jamee Guerra Valencia

**Affiliations:** 1Escuela de Nutrición y Dietética, Universidad Científica del Sur, Lima 15067, Peru; 100030483@cientifica.edu.pe (C.R.-E.); 100029211@cientifica.edu.pe (S.A.-F.); 2Instituto de Investigaciones en Ciencias Biomédicas (INICIB), Universidad Ricardo Palma, Lima 15039, Peru; ruben.espinoza@urp.edu.pe; 3Carrera de Nutrición y Dietética, Facultad de Ciencias de la Salud, Universidad San Ignacio de Loyola, Lima 15024, Peru; lorena.saavedra@usil.pe; 4Department of Medical Education, Herbert Wertheim College of Medicine, Florida International University, Miami, FL 33199, USA; nbarengo@fiu.edu; 5Escuela Superior de Medicina, Universidad Nacional de Mar del Plata, Mar del Plata 7600, Argentina; 6Facultad de Ciencias de la Salud, Universidad Privada del Norte, Lima 15314, Peru

**Keywords:** obesity, abdominal obesity, body fat distribution, adiposity, skinfold thickness, hypertension, type 2 diabetes mellitus, ROC curve, Latin America

## Abstract

The escalating prevalence of overall and abdominal obesity, particularly affecting Latin America, underscores the urgent need for accessible and cost-effective predictive methods to address the growing disease burden. This study assessed skinfold thicknesses’ predictive capacity for overall and abdominal obesity in Peruvian adults aged 30 or older over 5 years. Data from the PERU MIGRANT 5-year cohort study were analyzed, defining obesity using BMI and waist circumference. Receiver operating characteristic curves and area under the curve (AUC) with 95% confidence intervals (CI) were calculated. Adults aged ≥ 30 (*n* = 988) completed the study at baseline, with 47% male. A total of 682 participants were included for overall and abdominal obesity analysis. The 5-year prevalence values for overall and abdominal obesity were 26.7% and 26.6%, respectively. Subscapular skinfold (SS) best predicted overall obesity in men (AUC = 0.81, 95% CI: 0.75–0.88) and women (AUC = 0.77, 95% CI: 0.67–0.88). Regarding abdominal obesity, SS exhibited the highest AUC in men (AUC = 0.83, 95% CI: 0.77–0.89), while SS and the sum of trunk skinfolds showed the highest AUC in women. In secondary analysis excluding participants with type-2 diabetes mellitus (DM2) at baseline, SS significantly predicted DM2 development in men (AUC = 0.70, 95% CI: 0.58–0.83) and bicipital skinfold (BS) did in women (AUC = 0.73, 95% CI: 0.62–0.84). The findings highlight SS significance as an indicator of overall and abdominal obesity in both sexes among Peruvian adults. Additionally, SS, and BS offer robust predictive indicators for DM2.

## 1. Introduction

Obesity is a chronic pathological condition characterized by excessive fat accumulation leading to low-level chronic inflammation, and oxidative stress conditions [1]. Its escalating prevalence on a global scale has raised significant concerns for public health [2,3]. In fact, obesity contributed to approximately 160 million disability-adjusted life years (DALYs) by 2019, with Americas being the second most affected region [4]. Furthermore, excessive abdominal adiposity has been considered as an independent risk factor for different chronic diseases such as type 2 diabetes mellitus (DM2), hypertension (HTN), cardiovascular diseases, cardiovascular mortality, and total mortality [5,6]. The global trends of abdominal obesity are similar to those of overall obesity [7,8], with Latin America experiencing one of the most rapid growth rates [8]. In Peru, the prevalence of obesity among individuals over 15 years old was reported to be 25.8% in 2021 [9]. Abdominal obesity affects over 50% of the adult population in the country [10].

The accurate assessment of adiposity is crucial for understanding the impact of obesity on health. Although Body Mass Index (BMI) is the most frequently used method for assessing excessive adiposity both in clinical and research scenarios [11,12], alternative anthropometric measures, such as skinfold thickness, provide valuable information regarding the regionalization of adiposity [13]. Moreover, these measures offer distinct advantages, as they are noninvasive and relatively inexpensive compared to other techniques [13,14]. Skinfold thickness measurements are performed manually using a skinfold caliper, with commonly evaluated sites including the biceps, triceps, subscapular, and suprailiac regions [15].

While existing evidence substantiates the predictive capability of skinfold thickness measurements in assessing obesity [16,17,18], especially when compared to other anthropometric measures [16,19,20], further investigation is needed to enhance our comprehension of distinct obesity types, such as general or central obesity. This necessity is particularly important when considering the adult population in Latin America, where obesity rates are escalating at an alarming pace [3,8]. Notably, studies have indicated that Latin Americans tend to have a greater propensity for visceral adiposity accumulation compared to other ethnic groups [21,22]. Furthermore, it is important to acknowledge that most studies exploring the predictive potential of skinfold thickness measurements for various health outcomes have predominantly focused on children and adolescents, rather than adult cohorts [17,18,19]. Moreover, among the studies involving adult populations, a significant proportion have employed cross-sectional designs as opposed to prospective investigations [20,23,24].

Given the importance of accurate adiposity assessment and the need for a comprehensive understanding of obesity-related health risks, the primary objective of the present study was to evaluate the predictive capacity of skinfold thickness measurements for the development of overall obesity and abdominal obesity within a Peruvian cohort over a 5-year follow-up period. As a secondary objective, the study aimed to analyze the predictive capacity of skinfold thickness measurements for the development of hypertension and type 2 diabetes mellitus within the same 5-year follow-up cohort.

## 2. Materials and Methods

### 2.1. Study Design

This non-concurrent cohort study consisted of a secondary analysis of the PERU MIGRANT cohort study database developed by the CRONICAS Center. The analysis included data from the baseline assessment conducted in 2007–2008 and the 5-year follow-up assessment conducted in 2012–2013.

The study participants consisted of men and women aged 30 years or older without a history of mental illness or current pregnancy living in Peru. The recruitment of participants permanently living in selected rural and urban areas was performed using a single-stage random sampling method. Rural dwellers were selected from San Jose de Secce, Huanta, Ayacucho, while rural-to-urban migrants and urban dwellers were chosen from Las Pampas de San Juan de Miraflores. Migrants were defined as individuals who permanently lived in Las Pampas de San Juan de Miraflores at the time of the baseline assessment but were born in Ayacucho.

Enrollment at baseline was conducted through a single-stage random sampling technique. In 2007, a census was conducted for San Jose de Secce (rural area), and an updated local census was used for Lima (urban area). For both rural and urban areas, the sampling frame was based on a local census for the year 2000 [25]. The reported response rate at the baseline was 61.6% (989/1606) [25]. Participants were re-contacted and re-assessed at the original enrollment location for the 5-year follow-up assessment [25,26].

Detailed information regarding selection criteria, assessed variables, sample size, and participation rates have been previously published elsewhere [25,26].

### 2.2. Participant Selection

Only participants with relevant variables of interest were included in the present study. Normal-weight, overweight, and normal-waist-circumference participants were included, while those with overall obesity (defined as a BMI ≥ 30 kg/m^2^) and/or abdominal obesity (defined as waist circumference ≥ 102 cm in men and ≥ 88 cm in women) were excluded at baseline. Therefore, for the main objective, a total of 274 participants were excluded at baseline.

As a secondary objective, the diagnostic capacity of skinfold measurements for HTN and DM2 development was assessed. For the exclusion criteria at baseline, participants were handled separately for HTN and DM2. For HTN, individuals with systolic blood pressure (SBP) ≥ 140 mmHg or diastolic blood pressure (DBP) ≥ 90 mmHg, as well as those who self-reported being diagnosed with HTN or receiving HTN treatment at baseline, were excluded. For DM2, participants with a blood glucose level ≥ 126 mg/dL or who self-reported receiving DM2 treatment at baseline were excluded. A total of 207 and 23 participants were excluded for HTN and DM2, respectively.

### 2.3. Variables and Measurements

The main outcome variables were overall obesity, and abdominal obesity at the 5-year follow-up. Overall obesity was defined considering the World Health Organization (WHO) cut-off (BMI ≥ 30 kg/m^2^) [27], while abdominal obesity was defined following the ATP-III guidelines (waist circumference ≥ 102 cm in men and ≥ 88 cm in women) [28]. Both definitions have been further endorsed in the technical guidelines for adult nutritional assessment in Peru [29].

To test the robustness of the diagnostic accuracy results of skinfold thickness for overall and abdominal obesity, an additional criterion for defining abdominal obesity was also analyzed (waist circumference ≥ 94 cm in males and ≥ 92 cm in females). This criterion was based on a Latin American cut-off proposal for detecting a visceral adipose tissue ≥ 100 cm^2^ [30].

HTN and DM2 at the five-year follow-up were considered as secondary outcome variables. Hypertension was defined as SBP ≥ 140 mmHg or DBP ≥ 90 mmHg, self-reported HTN treatment, or physician-diagnosed HTN requiring prescribed medication. DM2 was defined as self-reported DM2 with physician-prescribed drugs.

The skinfold measurements examined in this study included the bicipital, tricipital, subscapular, and suprailiac areas, the sum of upper limb skinfolds (bicipital + tricipital), the sum of trunk skinfolds (subscapular + supra iliac), the upper limb-to-trunk skinfold ratio, the trunk-to-upper limb skinfold ratio, and the combined sum of the four site skinfolds. Each skinfold was measured in triplicate and recorded in millimeters, performed by a consistent field worker employing a HoltainTanner/Whitehouse Skinfold Caliper calibrated to the nearest 0.2 mm [31]. The standardized procedure for obtaining each skinfold measurement began with the secure grasp of the skinfold between the thumb and index finger of the observer’s left hand, ensuring the inclusion of all underlying adipose tissue within the fold. Subsequently, the skinfold caliper was positioned in the right hand, applied to the skinfold, and maintained at a consistent distance of approximately 1 cm from the fingers of the left hand, ensuring that only the caliper’s faces, and not the observer’s fingers, applied pressure to the fold. Following the precise placement of the caliper, the observer released the fingers of their right hand, allowing the instrument to exert its maximum pressure while sustaining the grip of the fold with their left hand. Measurements were recorded by reading the dial of the caliper to the nearest 0.2 mm. To enhance accuracy, the caliper was held in place for 2 s before each measurement was recorded.

Specific anatomical landmarks guided the placement of each skinfold measurement were as follows:

Tricipital: Positioned at the midpoint between the acromion and the olecranon on the posterior surface of the triceps muscle, this measurement was captured with the arm in a relaxed and slightly flexed position, with the palm facing forward, and approximately 1 cm above the designated measurement level.

Bicipital: Located on the anterior surface of the arm at the same height as the triceps skinfold.

Subscapular: Attained with the subject in an upright position, their back exposed, and arms at ease by their sides. The fold was taken at a slightly oblique angle, approximately 1 cm below the lower angle of the right scapula.

Supra iliac: Situated two centimeters above the left iliac crest, along the mid-axillary line.

All skinfolds were measured to complete a circuit, and this circuit of measurements was repeated three times. It was ensured that the same skinfold was not measured consecutively, preventing any potential bias or errors in the measurement. Finally, the average value from the three measurements of each skinfold was computed and employed for subsequent estimations.

Regarding the standardization process for anthropometric measurements, this was based on the approach of calculating the average of all observations conducted by the same observer. This was performed monthly before the study began, and each observer field worker was required to measure a minimum of 10 subjects in duplicate.

The covariates reported in this analysis were sex (male, female), age group (30 to 44, 45 to 59, and 60 or older), migration group (urban, rural, or migrant), current smoking status (yes/no), alcohol consumption (low/high), and physical activity level (high, moderate, and low). Smoking and alcohol consumption were assessed with an adapted version of the WHO STEPS questionnaire [32], while physical activity levels were defined according to the International Physical Activity Questionnaire (IPAQ) protocol [33]. The short form of IPAQ was used.

### 2.4. Statistical Analysis

The statistical analysis was conducted using STATA v17.0 software. Descriptive analyses were presented as absolute frequencies and percentages for categorical variables (sex, age group, migration group, current smoker, alcohol consumption, and physical activity level), while numerical variables were summarized as the median and interquartile range. The diagnostic performance of skinfold measurements was evaluated using receiver operating characteristic (ROC) curves, and the corresponding area under the curve (AUC) was calculated. The ROC curves were stratified according to sex. Comparisons among ROC curves were performed to determine the most appropriate test. Sensitivity (S), specificity (Sp), and positive and negative predictive values, as well as positive and negative likelihood ratios, were calculated. The optimal cut-off point for each skinfold measurement was determined using the Youden index.

### 2.5. Ethics Considerations

Ethics Committee approval was obtained from the Universidad Peruana Cayetano Heredia before the study commenced (Ethics Committee approval number 60014). The purpose of the study was thoroughly explained to each participant, and informed consent was subsequently obtained. Participants who were originally involved in the PERU MIGRANT 2007–2008 study were contacted between the years 2012 and 2013. Furthermore, the present study adhered to the ethical standards outlined in the Helsinki Declaration. Since the present study entailed an analysis of secondary, open-access data [34] there was no direct interaction with the individuals who were part of the original study. Therefore, no potential risks were posed to the participants.

## 3. Results

Out of the initially recruited participants at baseline, 33 participants died, leaving a total of 956 subjects with complete data at the follow-up.

Regarding overall and abdominal obesity outcomes, 274 were excluded after applying the exclusion criteria, resulting in a follow-up cohort of 682 participants (Figure 1). The inclusion of participants for these outcome assessments when using a different abdominal obesity criterion can be seen in Figure A1.

For HTN and DM2, 207 and 23 were excluded at baseline, respectively, resulting in follow-up cohorts of 749 and 933 participants for HTN and DM2 assessments (see Figure A2 and Figure A3).

At baseline enrollment of the original PERU MIGRANT study, 989 participants were assessed. Women comprised 52.78% of the original sample. A total of 84.53% were under 60 years old. A fifth of the study sample lived in rural areas. Most participants reported no current smoking (88.88%) or no consumption of high volumes of alcohol (92.82%). Almost half of the participants reported a high physical activity level. The stratified analysis by sex showed that all skinfold thickness median values significantly differed by sex, with women having greater skinfold values. The age group was equally distributed according to sex (*p* = 0.949). However, statistically significant differences in frequencies were observed for smoking, alcohol consumption, and physical activity levels (Table 1).

Finally, overall obesity and abdominal obesity prevalence levels at the 5-year follow-up were 26.7% and 26.6%, respectively.

At the 5-year follow-up, both abdominal obesity (*p* < 0.001) and overall obesity (*p* = 0.013) were different in men and women, with a higher proportion of women reporting abdominal obesity (36.52%) when compared with men (19.13%), whereas overall obesity was more frequent among men (30.36%) than among women (21.84%)

Diagnostic accuracy for overall- and abdominal-obesity development over a 5-year period is shown in Table 2. Subscapular skinfold thickness had the highest area under the curve (AUC) in both men (AUC = 0.81, 95% CI: 0.75–0.88) and women (AUC = 0.77, 95% CI: 0.66–0.88) for overall obesity. The optimal cut-off values were 18.63 mm (sensitivity = 77.70%, specificity = 72.40%) for men and 21.77 mm (sensitivity = 74.80%, specificity = 68.80%) for women. Similarly, for the development of abdominal obesity over a 5-year period, subscapular skinfold thickness had the highest AUC in men (AUC = 0.83, 95% CI: 0.77–0.89) with a cut-off value of 17.70 mm (sensitivity = 71.90%, specificity = 85.70%). In women, the sum of subscapular + suprailiac skinfold thicknesses showed the highest AUC of 0.78 (95% CI: 0.72–0.83). The optimal cut-off value for subscapular + suprailiac skinfold thicknesses was 43.00 mm (sensitivity = 68.90%, specificity = 73.80%). Notably, the subscapular skinfold performed as well as the sum of the trunk skinfold, with a slightly lower specificity.

Additional analysis for overall and abdominal obesity with a different abdominal obesity criterion was performed (Table A1). Similarly to the reported results in Table 2, when defining abdominal obesity as waist circumference ≥ 94 cm in males and ≥92 cm in females, the subscapular skinfold showed the highest AUC for men (AUC = 0.85, 95% CI: 0.81–0.90). However, in women, the AUC and sensitivity increased when adding the suprailiac skinfold to the subscapular site (AUC = 0.84, 95% CI: 0.80–0.87; sensitivity = 81.20%, specificity = 72.9%).

To further explore the predictive capacity of skinfold thickness, a secondary analysis was conducted by excluding participants with hypertension and type 2 diabetes mellitus at baseline. For this purpose, 207 HTN and 23 DM2 participants were excluded at baseline. For HTN development, subscapular skinfold thickness demonstrated the highest AUC in men (AUC = 0.67, 95% CI: 0.48–0.86) with a cut-off value of 17.6 mm (sensitivity = 64.50%, specificity = 71.40%). Conversely, for women, the bicipital skinfold had the highest AUC at 0.58 (95% CI: 0.39–0.78) with a cut-off value of 12.3 mm (sensitivity = 54.80%, specificity = 75.00%).

Similar trends were observed in the context of type 2 diabetes mellitus development. Among men, subscapular skinfold thickness showed the highest AUC (AUC = 0.70, 95% CI: 0.58–0.83) with a cut-off value of 17.80 mm (sensitivity = 66.30%, specificity = 66.70%). Worth noting is that when the suprailiac skinfold measurement was added to the subscapular measurement, the AUC remained unchanged; however, specificity increased to 81.80%, albeit at the cost of reduced sensitivity. In women, the bicipital skinfold had the highest AUC of 0.73 (95% CI: 0.62–0.84) with a cut-off value of 16.90 mm (sensitivity = 73.10%, specificity = 64.30%). Detailed results of the predictive capacity analysis can be found in Table 3 and Table 4.

## 4. Discussion

The present study aimed to assess the diagnostic capabilities of skinfold thickness measurements for overall obesity and abdominal obesity development. The results indicated that the subscapular skinfold measurement showed the best diagnostic capacity for overall obesity development in men and women. Conversely, abdominal obesity was best predicted by the subscapular skinfold in men, while for women, a combination of measurements, including subscapular skinfold, the sum of trunk skinfolds, and the sum of skinfolds across four sites, provided the most accurate predictions.

All skinfold thicknesses were found to be statistically significantly higher in women when compared to men. This finding is not unexpected as sexual dimorphism in the subcutaneous adipose distribution has been reported as early as puberty onset [35,36], which can be tracked into adulthood [23,35,37]. In the present study, subscapular skinfold thickness exhibited the highest predictive capacity for overall obesity development in a 5-year follow-up period. This finding is in line with previous studies reporting that the subscapular skinfold is a sensitive predictor for adult fatness accretion since adolescence [37] and for central fat mass among adults [20]. Furthermore, Ramírez-Vélez [18] reported an AUC of 0.72 in women and 0.78 in men for the subscapular skinfold prediction of overall obesity. Similarly, Myrtaj [17] found that both the subscapular skinfold and the sum of four skinfolds had a high predictive power for obesity in adolescents in Macedonia (women: AUC = 0.865 and 0.907, men: AUC = 0.928 and 0.941). Although both Ramírez-Vélez and Myrtaj’s studies were performed in pediatric populations, this should not neglect the predictive capacity of skinfold thickness, as body fatness may vary up to 5% when the subscapular skinfold is added to BMI and waist circumference into regression models in the adult population [20].

Regarding abdominal obesity development, subscapular skinfold was the best of all skinfold predictors among men, whereas in women both subscapular and the sum of trunk skinfold showed the best performance. Furthermore, the addition of the suprailiac skinfold exhibited a slightly higher specificity than the subscapular one alone in women. Additionally, when a stricter abdominal obesity definition was used [30], skinfold diagnostic performance remained stable among men, whereas in women, subscapular skinfold and the sum of trunk skinfold increased their AUC and sensitivity. Potential explanations for these findings are related to the aging pattern of body fat redistribution with a characterized reduction in appendicular fat and an increase in trunk fat [38], which may explain why adding the suprailiac to subscapular skinfold slightly increased the AUC both when using the ATP-III and a stricter abdominal obesity criteria. In line with this, a previous study reported that aging increases the prevalence of abdominal obesity by several times among men and to a greater extent than in women, when Peruvian normal-weight individuals were assessed [39]. On the other hand, while aging also contributes to an increase in trunk adiposity accretion among women [40], it is acknowledged that they tend to exhibit higher peripheral-to-central subcutaneous adiposity [20]. However, it has also been reported that patterns of adipose distribution in women exhibit greater variability among ethnic groups than in men [41]. Furthermore, this may be explained when considering the ethnicity of the study population as a previous study found that among five different Latin American countries, the genomics of Peruvian and Mexican populations were positively associated with an abdominal fat distribution more severely present in the female sex [21].

While the sum of trunk skinfold and all-four-site skinfold measurements showed a similar performance to subscapular skinfold for identifying overall and abdominal obesity in women, it is worth noting that multiple-site skinfold assessment can be time-consuming. Additionally, our study did not observe a significant enhancement in diagnostic accuracy with the inclusion of additional skinfold measurements. Consequently, subscapular skinfold measurement appears to be a clinically useful option for identifying overall and abdominal obesity.

Given the reported ethnic variability in body fat distribution [18,21], it is important to establish appropriate cut-off values specific to the Peruvian population. In the present study, we propose specific cut-off points for predicting overall obesity and central obesity based on subscapular skinfold measurements. For women, a cut-off point of 21.77 mm for the subscapular skinfold is suggested for overall obesity prediction, and an 18.93 mm cut-off point is recommended for central obesity prediction. In men, a cut-off point of 18.63 mm for the subscapular skinfold is proposed for overall obesity prediction. For central obesity prediction in men, we recommend a cut-off point of 17.70 mm for the subscapular skinfold. Although limited data regarding skinfold thickness measurements in the Peruvian adult population have been published, studies involving Caucasian adults reported mean subscapular skinfold measurements of 15 mm and 13 mm for women and men, respectively [20]. Given the more pronounced central adiposity observed in Latin American individuals compared to Caucasians, it can be expected that a significant proportion of Peruvian adults would exceed the recommended cut-off values from this study. Nevertheless, further research in this field is warranted, as subcutaneous adiposity, measured by skinfold thickness, has been reported to be lower in the Peruvian Andean population compared to the population of the United States [42].

As a secondary objective, the present study aimed to analyze the predictive capacity of skinfold thickness for HTN and DM2 development. It was observed that subcutaneous skinfolds have a low accuracy as diagnostic tests for predicting hypertension showed an AUC lower than 0.70 and without statistical significance. Similarly, Ali [43] reported a low but statistically significant predictive capacity of the subscapular skinfold for high systolic blood pressure in male university students in Egypt (AUC = 0.645). Additionally, other studies conducted in the pediatric population have reported an AUC for skinfold thicknesses lower than 0.70 [44,45]. Furthermore, other anthropometric measures, such as body mass index, waist circumference, waist-to-hip ratio, and waist-to-height ratio, have also shown low predictive values for hypertension diagnosis (AUC < 0.6) in both men and women [44,46,47,48].

On the contrary, the present study found that the subscapular skinfold showed a useful predictive capacity for DM2 in men (AUC = 0.70), while the bicipital skinfold outperformed it in women (AUC = 0.73). Furthermore, among men, an increase in specificity was shown when the suprailiac skinfold was added to subscapular measurement. Although research in this field is limited, our findings concur with a previous study conducted on young Mexican women [49], which found a good predictive value of the sum of skinfold thickness for pre-diabetes development on a 5-year follow-up period. Furthermore, this study showed no statistically significant differences when comparing the sum of skinfolds (bicipital, tricipital, subscapular, and supraspinal) with classical anthropometric measurements such as BMI and waist circumference predictive capacity [49]. Different studies conducted on adults in Iran [50] and on older adults in China [51] found that none of these indicators had a good predictive capacity (AUC < 0.70). However, Gordillo [52] found good predictive capacity for waist circumference (AUC = 0.747) and the waist-to-height ratio (AUC = 0.737) in Ecuadorian adults. Similarly, Machoene [53] revealed that waist circumference, the waist-to-hip ratio, and the waist-to-height ratio had very good predictive values (AUC = 0.804, 0.802, and 0.806, respectively) in South African adult men. These mixed results may be explained by considering that negative health outcomes may appear at different thresholds of commonly used anthropometric measurements depending on the country of origin and ethnic group [49]. Therefore, the present study recommends a cut-off for the subscapular skinfold to predict DM2 development in men of 17.80 mm and a cut-off point for the bicipital skinfold of 16.90 mm in women.

It is worth highlighting the potential of skinfold thickness assessment in enhancing DM2 surveillance and comorbidity screening. Given that DM2 and its complications ranked among the top ten leading causes of disability-adjusted life years (DALYs) globally during the 2007–2017 period [54] and that age-standardized DALYs due to DM2 are particularly elevated in Andean Latin America [55], integrating skinfold thickness assessment could prove to be instrumental in the early detection and management of this prevalent health concern.

This study has some limitations and strengths that should be considered. Participants included in the study came from rural, urban, and migrant settings, making it important to recognize that the findings may not be directly applicable to the broader Peruvian population. In line with this, a previous study conducted in the Andean Peruvian population found about half of the subscapular values in the present study [42]. Therefore, it is possible that this population has been underrepresented in this study. Additionally, due to limitations in sample size, we conducted ROC analysis stratified by sex rather than considering variations in age distribution.

The technical error of measurement for skinfold thickness was not reported for the PERU MIGRANT study. Nevertheless, anthropometric measurements were recorded in triplicate and by the same fieldworker [25], therefore minimizing intra-evaluator error [56].

The average of triplicate skinfold measures was used instead of the median value as individual values were not available in the free online database, thus preventing us from calculating the median values. As there is a possibility that the average value may be skewed if one of the readings largely differed from the other two, caution is suggested when interpreting the proposed cut-offs. However, anthropometric measurements followed a standardization process based on the approach of calculating the average of all observations conducted by the same observer, potentially minimizing precision errors. Regarding assessed adiposity outcomes, both overall and abdominal obesity were defined based on anthropometric measures rather than in more objective methods such as DEXA or others. However, internationally accepted cut-offs for BMI and waist circumference were used to define overall and abdominal obesity. These criteria are also endorsed by the Peruvian technical guidelines for nutritional assessment [29].

Similarly, DM2 outcome was defined based on self-report diagnosis or physician-prescribed drugs, for which classification bias may exist. Nonetheless, even considering the under-diagnosis of T2DM rates [57], our findings remain robust, as AUC, sensitivity, and specificity are not contingent on prevalence.

Despite these limitations, it is worth highlighting the strengths of this study, such as its prospective nature. Furthermore, our findings were confirmed by a second criterion for defining abdominal obesity. Additionally, no previous studies in the Latin America region have analyzed the predictive values and cut-offs for maximizing sensitivity and specificity in both adult men and women populations.

## 5. Conclusions

In summary, the present study provides compelling evidence supporting the diagnostic potential of subscapular skinfold measurements in identifying both overall and abdominal obesity. Moreover, our findings highlight the significance of subscapular and bicipital skinfold measurements as robust predictors for the development of type 2 diabetes mellitus in men and women, respectively. Considering their cost-effectiveness, non-invasiveness, and ability to provide valuable insights into regional adiposity, skinfold thickness measurements represent invaluable tools for predicting metabolic risks and assessing health conditions associated with adiposity. Given the implications of these findings, we strongly advocate for the routine incorporation of skinfold assessments into clinical practice, as they have the potential to enhance obesity surveillance and comorbidity screening. However, to ensure broader applicability and generalizability, further investigations are warranted, particularly in diverse populations. Continued research endeavors will serve to validate and refine the findings, thus advancing our understanding of the role of skinfold thickness measurements in assessing adiposity-related health risks.

## Figures and Tables

**Figure 1 ijerph-20-07089-f001:**
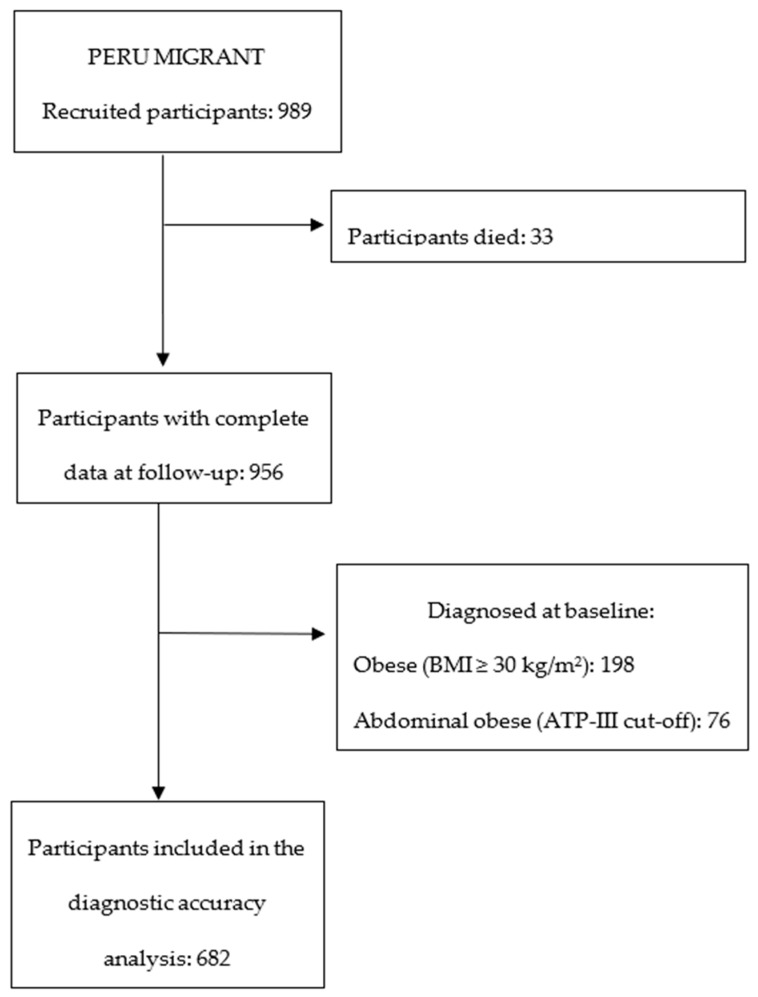
Flow diagram of the participant inclusion in the overall- and abdominal-obesity diagnostic accuracy analysis.

**Table 1 ijerph-20-07089-t001:** Characteristics of the sample at baseline by sex, PERU MIGRANT Cohort Study, 2007–2008.

	Total Sample	Female	Male	*p*-Value *
Characteristics	*n* (%)	*n* (%)	*n* (%)	
**Age group**				0.949
30 to 44	429 (43.38)	228 (53.15)	201 (46.85)	
45 to 59	407 (41.15)	215 (52.83)	192 (47.17)	
60 or older	153 (15.47)	79 (51.63)	74 (48.37)	
**Migration group**				0.95
Rural	201 (20.32)	106 (52.74)	95 (47.26)	
Migrant	589 (59.56)	309 (52.46)	280 (47.54)	
Urban	199 (20.12)	107 (53.77)	92 (46.23)	
**Smoking**				<0.001
No	879 (88.88)	503 (57.22)	376 (42.78)	
Yes	110 (11.12)	19 (17.27)	91 (82.73)	
**Alcohol consumption**				<0.001
Low	918 (92.82)	516 (56.21)	402 (43.79)	
High	71 (7.18)	6 (8.45)	65 (91.55)	
**Physical Activity level**				<0.001
Low	255 (25.99)	137 (53.73)	118 (46.27)	
Moderate	286 (29.15)	180 (62.94)	106 (37.06)	
High	440 (44.85)	201 (45.68)	239 (54.32)	
**Skinfolds, BMI, and Waist circumference ^¥ ¶^**				
**Bicipital**	7.8 (5.0–13.6)	11.7 (7.5–18.3)	5.3 (4.0–7.4)	<0.001
**Tricipital**	18.9 (11.2–31.2)	27.7 (18.5–36.6)	11.7 (8.5–17.5)	<0.001
**Suprailiac**	25.2 (16.3–31.8)	27.6 (19.2–33.7)	22.1 (14.1–29.3)	<0.001
**Subscapular**	18.6 (13.4–25.7)	21.8 (16.0–28.2)	16 (11.3–20.5)	<0.001
**Bicipital + Tricipital**	27.1 (16.7–44.9)	40.3 (26.9–54.7)	17.3 (12.6–25.7)	<0.001
**Subscapular + Suprailiac**	44.3 (30.6–57.1)	50.8 (35.5–60.6)	38.3 (25.8–49.5)	<0.001
**Upper limb-to-trunk skinfold ratio**	0.68 (0.51–0.89)	0.85 (0.71–1.01)	0.50 (0.43–0.60)	<0.001
**Trunk-to-upper limb skinfold ratio**	1.46 (1.11–1.95)	1.16 (0.98–1.39)	1.98 (1.64–2.33)	<0.001
**Sum of skinfolds**	71.8 (49.7–101.9)	92.3 (63.2–115.0)	57.5 (39.3–76.6)	<0.001
**BMI**	25.9 (23.2–29.0)	26.8 (23.6–30.4)	25.2 (22.8–27.6)	<0.001
**Waist circumference**	86.1 (78.5–94.1)	84.9 (76.7–94.5)	86.3 (80.2–93.5)	0.017

* Assessed with chi-squared test of independence. ^¥^ Median (interquartile range). ^¶^ Assessed with Mann–Whitney test.

**Table 2 ijerph-20-07089-t002:** Diagnostic values of the skinfold thickness for overall and abdominal obesity, stratified by sex.

	Cut-Off	AUC	95% CI–AUC	S (%)	Sp (%)	NPV (%)	PPV (%)	LR+ (%)	LR− (%)
**Male**									
**Overall obesity**									
Bicipital	5.90	0.72	(0.64–0.79)	67.80	62.10	95.90	12.90	178.80	51.80
Tricipital	10.90	0.70	(0.61–0.78)	51.10	79.30	97.00	11.10	246.80	61.70
Supra-iliac	22.03	0.75	(0.67–0.82)	58.20	89.30	98.60	13.70	543.60	46.80
Subscapular	18.63	0.81	(0.75–0.88)	77.70	72.40	97.30	20.00	281.50	30.90
Sum of skinfold	58.03	0.75	(0.68–0.83)	60.90	78.60	97.40	13.00	284.20	49.80
Bicipital + Tricipital	18.40	0.70	(0.62–0.78)	63.30	62.10	95.60	11.50	172.50	59.10
Subscapular + Supra-iliac	41.97	0.77	(0.69–0.84)	69.10	72.40	97.00	15.30	224.10	42.60
Upper limb-to-trunk skinfold ratio	0.47	0.56	(0.45–0.68)	62.50	55.20	94.80	10.20	166.70	68.00
Trunk-to-upper limb skinfold ratio	2.15	0.56	(0.45–0.68)	62.50	55.20	94.80	10.20	166.70	68.00
**Abdominal obesity**									
Bicipital	6.17	0.67	(0.57–0.76)	70.10	57.10	96.80	9.40	163.50	52.40
Tricipital	11.30	0.67	(0.57–0.76)	53.90	76.20	97.60	8.30	226.40	60.50
Supra-iliac	25.17	0.72	(0.61–0.82)	68.20	70.00	97.80	10.30	227.40	45.40
Subscapular	17.70	0.83	(0.77–0.89)	71.90	85.70	98.90	14.30	503.10	32.80
Sum of skinfold	63.37	0.74	(0.65–0.83)	71.90	75.00	98.20	12.20	287.50	37.50
Bicipital + Tricipital	15.83	0.67	(0.57–0.76)	50.30	85.70	98.50	8.60	101.00	58.00
Subscapular + Suprailiac	44.30	0.75	(0.65–0.85)	73.70	71.40	97.90	12.90	280.20	36.80
Upper limb-to-trunk skinfold ratio	0.45	0.64	(0.51–0.77)	67.40	66.70	97.40	10.10	207.20	48.80
Trunk-to-upper limb skinfold ratio	2.21	0.64	(0.51–0.77)	67.40	66.70	97.40	10.10	207.20	48.80
**Female**									
**Overall obesity**									
Bicipital	10.43	0.74	(0.63–0.84)	67.80	68.80	97.50	10.70	217.10	46.80
Tricipital	25.30	0.74	(0.62–0.86)	62.90	75.00	97.80	10.20	251.70	49.40
Suprailiac	25.67	0.69	(0.57–0.82)	65.80	68.80	97.40	10.20	210.70	49.70
Subscapular	21.77	0.77	(0.66–0.88)	74.80	68.80	97.70	13.30	239.40	36.60
Sum of skinfold	81.20	0.75	(0.63–0.86)	62.70	75.00	97.80	10.20	250.70	49.80
Bicipital + Tricipital	35.30	0.74	(0.62–0.85)	65.00	75.00	97.90	10.70	186.00	46.60
Subscapular + Suprailiac	53.73	0.74	(0.61–0.86)	81.50	62.50	97.50	15.90	439.60	29.70
Upper limb-to-trunk skinfold ratio	0.97	0.56	(0.40–0.72)	72.40	56.30	96.70	10.20	262.00	49.10
Trunk-to-upper limb skinfold ratio	1.03	0.56	(0.40–0.72)	72.40	56.30	96.70	10.20	262.00	49.10
**Abdominal obesity**									
Bicipital	9.03	0.74	(0.68–0.80)	63.10	68.80	84.80	40.10	201.80	53.70
Tricipital	23.43	0.74	(0.68–0.80)	66.70	66.30	84.60	41.70	197.50	50.30
Suprailiac	23.90	0.75	(0.70–0.81)	66.40	72.50	86.90	43.90	241.30	46.40
Subscapular	18.93	0.77	(0.71–0.82)	68.50	71.30	86.90	44.90	238.20	44.30
Sum of skinfold	80.17	0.77	(0.72–0.83)	69.10	67.50	85.40	44.30	212.60	45.80
Bicipital + Tricipital	30.93	0.75	(0.69–0.80)	62.20	72.50	86.30	40.80	164.30	52.20
Subscapular + Suprailiac	43.00	0.78	(0.72–0.83)	68.90	73.80	87.90	46.10	221.70	42.10
Upper limb-to-trunk skinfold ratio	0.83	0.52	(0.45–0.60)	53.60	52.50	75.80	29.00	115.50	88.40
Trunk-to-upper limb skinfold ratio	1.21	0.52	(0.45–0.60)	53.60	52.50	75.80	29.00	115.50	88.40

AUC: area under the curve, S: sensitivity, Sp: specificity, PPV: positive predictive value, NPV: negative predictive value, LR+: positive likelihood ratio, LR−: likelihood ratio negative, 95% CI: 95% confidence interval.

**Table 3 ijerph-20-07089-t003:** Diagnostic values of the skinfold thickness for hypertension development, stratified by sex.

	Cut-Off	AUC	95% CI–AUC	S (%)	Sp (%)	NPV (%)	PPV (%)	LR+ (%)	LR− (%)
**Male**									
Bicipital	6.23	0.65	(0.48–0.82)	64.20	71.40	99.20	3.60	224.80	50.10
Tricipital	10.97	0.42	(0.23–0.60)	53.70	57.10	98.50	2.30	125.20	81.10
Suprailiac	22.17	0.60	(0.44–0.77)	51.40	71.40	99.00	2.70	179.80	68.10
Subscapular	17.60	0.67	(0.48–0.86)	64.50	71.40	99.20	3.70	225.70	49.70
Sum of skinfold	58.70	0.62	(0.45–0.79)	56.00	71.40	99.00	3.00	195.90	61.60
Bicipital + Tricipital	16.97	0.41	(0.23–0.59)	51.10	57.10	98.40	2.20	104.60	85.50
Subscapular + Suprailiac	41.83	0.64	(0.48–0.81)	59.80	71.40	99.10	3.40	149.00	56.20
Upper limb-to-trunk skinfold ratio	0.47	0.60	(0.41–0.79)	60.10	71.40	99.10	3.40	150.70	55.80
Trunk-to-upper limb skinfold ratio	2.13	0.60	(0.41–0.79)	60.10	71.40	99.10	3.40	150.70	55.80
**Female**									
Bicipital	12.30	0.58	(0.39–0.78)	54.80	75.00	99.10	3.30	219.00	60.30
Tricipital	32.00	0.50	(0.26–0.74)	62.50	50.00	98.40	2.70	124.90	75.10
Suprailiac	34.00	0.51	(0.27–0.75)	74.70	50.00	98.60	3.90	149.50	50.50
Subscapular	18.73	0.51	(0.29–0.74)	62.20	62.50	98.80	3.30	165.90	60.50
Sum of skinfold	103.73	0.51	(0.29–0.73)	66.20	50.00	98.50	3.00	132.50	67.50
Bicipital + Tricipital	44.23	0.52	(0.29–0.74)	58.30	50.00	98.20	2.40	140.00	83.30
Subscapular + Suprailiac	41.10	0.50	(0.26–0.75)	69.30	50.00	98.50	3.30	225.40	61.50
Upper limb-to-trunk skinfold ratio	0.82	0.54	(0.35–0.74)	46.40	75.00	98.90	2.80	86.40	71.50
Trunk-to-upper limb skinfold ratio	1.23	0.54	(0.35–0.74)	46.40	75.00	98.90	2.80	86.40	71.50

AUC: area under the curve, S: sensitivity, Sp: specificity, PPV: positive predictive value, NPV: negative predictive value, LR+: positive likelihood ratio, LR−: likelihood ratio negative, 95% CI: 95% confidence interval.

**Table 4 ijerph-20-07089-t004:** Diagnostic values of the skinfold thickness for type 2 diabetes mellitus development, stratified by sex.

	Cut-Off	AUC	95% CI–AUC	S (%)	Sp (%)	NPV (%)	PPV (%)	LR+ (%)	LR− (%)
**Male**									
Bicipital	5.30	0.66	(0.53–0.80)	51.50	75.00	98.10	5.70	205.80	64.70
Tricipital	16.43	0.64	(0.49–0.80)	74.80	50.00	97.50	7.10	149.50	50.50
Suprailiac	22.70	0.58	(0.44–0.72)	53.40	66.70	97.60	5.30	160.20	69.90
Subscapular	17.80	0.70	(0.58–0.83)	66.30	66.70	98.10	7.10	199.00	50.50
Sum of skinfold	58.63	0.65	(0.50–0.79)	56.00	66.70	97.70	5.60	168.00	66.00
Bicipital + Tricipital	19.17	0.69	(0.53–0.84)	58.90	72.70	98.80	4.40	143.00	56.60
Subscapular + Suprailiac	41.97	0.70	(0.58–0.81)	59.60	81.80	99.20	5.10	147.30	49.40
Upper limb-to-trunk skinfold ratio	0.50	0.44	(0.24–0.65)	51.00	54.50	97.70	2.80	103.90	89.90
Trunk-to-upper limb skinfold ratio	2.01	0.44	(0.24–0.65)	51.00	54.50	97.70	2.80	103.90	89.90
**Female**									
Bicipital	16.90	0.73	(0.62–0.84)	73.10	64.30	98.10	8.70	204.60	41.90
Tricipital	32.00	0.67	(0.54–0.79)	61.90	71.40	98.20	7.00	216.60	53.40
Suprailiac	28.30	0.65	(0.51–0.79)	53.70	64.30	97.40	5.30	150.50	72.00
Subscapular	28.97	0.72	(0.59–0.85)	80.20	57.10	97.90	10.40	187.20	34.60
Sum of skinfold	114.13	0.70	(0.57–0.83)	75.60	57.10	97.80	8.60	176.30	42.70
Bicipital + Tricipital	40.37	0.59	(0.42–0.77)	50.50	60.00	98.40	2.40	102.10	82.50
Subscapular + Suprailiac	53.47	0.63	(0.46–0.81)	56.90	60.00	98.60	2.80	131.90	71.90
Upper limb-to-trunk skinfold ratio	0.76	0.55	(0.38–0.72)	66.30	50.00	98.50	3.00	197.00	67.40
Trunk-to-upper limb skinfold ratio	1.32	0.55	(0.38–0.72)	66.30	50.00	98.50	3.00	197.00	67.40

AUC: area under the curve, S: sensitivity, Sp: specificity, PPV: positive predictive value, NPV: negative predictive value, LR+: positive likelihood ratio, LR−: likelihood ratio negative, 95% CI: 95% confidence interval.

## Data Availability

The data from the PERU MIGRANT study are publicly accessible at https://figshare.com/articles/dataset/PERU_MIGRANT_Study_Baseline_and_5yr_follow-up_dataset/4832612 (accessed on 23 April 2023).

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
