# Peer review of "Diagnostic Performance of the Measurement of Skinfold Thickness for Abdominal and Overall Obesity in the Peruvian Population: A 5-Year Cohort Analysis"

_ijerph, 2023, doi:10.3390/ijerph20237089_

Round 1

Reviewer 1 Report

This manuscript reports tests of the relationships between baseline skinfolds and 5 year later obesity, hypertension and DM2. New cut off values for skinfolds for the Peruvian population are proposed.

A major limitation of this study is the use of BMI as the index of obesity. This is a serious limitation.

Measuring skinfolds is time consuming and fraught with error. While the sum of skinfolds did best for women, the subscapular skinfold didn’t do bad, and would be less challenging than assessing 4 sites? If the authors are proposing use of skinfolds in clinical settings, seems subscapula readings should be endorsed. If the investigators are just interested in research use that should be stated.

Substantial information is missing. What % of the contactees agreed to participate? How representative was the sample of the targeted population (from gov’t collected data)?

Please provide more information on the skinfolds: how were the sites detected? How were the calipers employed? What was the sequence of skinfold site assessment?

Has the WHO STEPS been validated? Please provide a reference.

How many items in the IPAQ were used? Has this been validated?

Were incentives offered?

There were no drop outs? That is unheard of and needs more explanation.

Table 1: The CI should be values around the mean, but are not? Please explain or revise?

Author Response

Please see the attachment. A revised version of the manuscript was uploaded.

In detail responses to observations are as follows:

Point 1: A major limitation of this study is the use of BMI as the index of obesity. This is a serious limitation.

Response 1: We appreciate the reviewer's consideration of the limitations associated with using BMI as an index of obesity. While we acknowledge that BMI has its shortcomings as a comprehensive measure of obesity, it remains a universally recognized index for obesity definition, with a well-established cutoff of ≥ 30 kg/m² [1]. Furthermore, it's noteworthy that current Peruvian guidelines for obesity assessment also endorse the use of BMI as the criteria index, reinforcing its relevance within our specific study context [2]. It is essential to recognize that BMI, with a cutoff of ≥ 30 kg/m², has demonstrated strengths in certain aspects of obesity diagnosis as reported by a systematic review that this threshold exhibits near-perfect specificity and yields high post-test probabilities when used as a diagnostic criterion [3]. While it may exhibit lower sensitivity when compared to more sophisticated body composition techniques, these findings underscore the practical utility of BMI in particular settings. Our choice of BMI as an obesity index was grounded in its widespread acceptance, alignment with local guidelines, and its ability to effectively identify individuals at a higher risk of obesity-related health complications. However, we also acknowledge its limitations, particularly in capturing variations in body composition.

In the limitations section we have added a sentence regarding BMI use as obesity criteria endorsed by current Peruvian guidelines.

It reads: “These criteria are also endorsed by the Peruvian technical guidelines for nutritional assessment.

Point 2: Measuring skinfolds is time consuming and fraught with error. While the sum of skinfolds did best for women, the subscapular skinfold didn’t do bad, and would be less challenging than assessing 4 sites? If the authors are proposing use of skinfolds in clinical settings, seems subscapula readings should be endorsed. If the investigators are just interested in research use that should be stated.

Response 2: We agree with the reviewer. The previous sentence “It is worth considering that while the subscapular skinfold showed better results than the sum of skinfolds overall, measuring multiple location skinfolds could minimize the risk of possible abnormal fat distribution in many subjects” has been deleted.

Now it reads: “While the sum of trunk skinfold and all-four-site skinfold measurements showed similar performance to subscapular skinfold for identifying overall and abdominal obesity in women, it's worth noting that multiple-site skinfold assessment can be time-consuming. Additionally, our study did not observe a significant enhancement in diagnostic accuracy with the inclusion of additional skinfold measurements. Consequently, subscapular skinfold measurement appears to be a clinically useful option for identifying overall and abdominal obesity.”

Point 3: Substantial information is missing. What % of the contactees agreed to participate? How representative was the sample of the targeted population (from gov’t collected data)?

Response 3: We have added two sentences regarding the response rate at baseline and the sampling frame in the methods section (study design).

It reads: “For both, rural and urban areas, the sampling frame was based on a local census for the year 2000. The reported response rate at the baseline was 61.6% (989/1606)

Regarding representativeness, as a local frame sample was used, the resulting sample was representative of the analyzed variables. In support of this, sociodemographic and lifestyle characteristics previously reported in the PERU MIGRANT study [4] comport with national Peruvian Demographic and Health Surveys [5]

Point 4: Please provide more information on the skinfolds: how were the sites detected? How were the calipers employed? What was the sequence of skinfold site assessment?

Response 4: A full description has been added to the methods section (variables and measurements).

Now it reads: “The standardized procedure for obtaining each skinfold measurement began with the secure grasp of the skinfold between the thumb and index finger of the observer's left hand, ensuring the inclusion of all underlying adipose tissue within the fold. Subsequently, the skinfold caliper was positioned in the right hand, applied to the skinfold, and maintained at a consistent distance of approximately 1 cm from the fingers of the left hand, ensuring that only the caliper's faces, and not the observer's fingers, applied pressure to the fold. Following the precise placement of the caliper, the observer re-leased the fingers of the right hand, allowing the instrument to exert its maximum pressure while sustaining the grip of the fold with the left hand. Measurements were recorded by reading the dial of the caliper to the nearest 0.2 mm. To enhance accuracy, the caliper was held in place for 2 seconds before each measurement was recorded.

Specific anatomical landmarks guided the placement of each skinfold measurement as follows:

Tricipital: Positioned at the midpoint between the acromion and the olecranon on the posterior surface of the triceps muscle, this measurement was captured with the arm in a relaxed and slightly flexed position, with the palm facing forward, and approximately 1 cm above the designated measurement level.

Bicipital: Located on the anterior surface of the arm at the same height as the triceps skinfold.

Subscapular: Attained with the subject in an upright position, their back exposed, and arms at ease by their sides. The fold was taken at a slightly oblique angle, approximately 1 centimeter below the lower angle of the right scapula.

Supra iliac: Situated two centimeters above the left iliac crest, along the mid-axillary line.

All skinfolds were measured to complete a circuit, and this circuit of measurements was repeated three times. It was ensured that the same skinfold was not measured consecutively, preventing any potential bias or errors in measurement. Finally, the average value from the three measurements of each skinfold was computed and employed for subsequent estimations.

Regarding the standardization process for anthropometric measurements, this was based on the approach of calculating the average of all observations conducted by the same observer. This was performed monthly before the study began, and each observer field worker was required to measure a minimum of 10 subjects in duplicate.

Point 5: Has the WHO STEPS been validated? Please provide a reference.

Response 5: Reference has been added after mentioning the tool. The WHO STEPS questionnaire is a standardized and widely recognized framework for assessing non-communicable disease risk factors. It has been implemented in 122 countries across the WHO regions [6], underlining its global acceptance and utility for monitoring the variables of interest in our study. We believe that the WHO STEPS questionnaire, as a well-established and internationally adopted tool, is highly suitable for reporting the variables of interest in our study, and we have incorporated this valuable feedback into our revised manuscript.

Point 6: How many items in the IPAQ were used? Has this been validated?

Response 6:

In our study, we utilized the short form of IPAQ, which consists of 7 items. To clarify this in the manuscript, we have added a sentence that states: "The short form of IPAQ was used."

Furthermore, it's important to note that the IPAQ has undergone validation processes in various contexts. To provide the reviewer with the relevant source for validation details, we would like to direct them to the following references: Craig, C. L., Marshall, A. L., Sjöström, M., Bauman, A. E., Booth, M. L., Ainsworth, B. E., Pratt, M., Ekelund, U., Yngve, A., Sallis, J. F., & Oja, P. (2003). International physical activity questionnaire: 12-country reliability and validity. Medicine and science in sports and exercise, 35(8), 1381–1395  And Bauman, A., Ainsworth, B. E., Bull, F., Craig, C. L., Hagströmer, M., Sallis, J. F., Pratt, M., & Sjöström, M. (2009). Progress and Pitfalls in the Use of the International Physical Activity Questionnaire (IPAQ) for Adult Physical Activity Surveillance. Journal of Physical Activity and Health, 6(s1), S5-S8. Retrieved Sep 5, 2023, from https://doi.org/10.1123/jpah.6.s1.s5

Point 7: Were incentives offered?

Response 7: No monetary incentives were offered. This was clearly stated in the consent form. This can be accessed at: https://bmccardiovascdisord.biomedcentral.com/articles/10.1186/1471-2261-9-23 (see for supplementary materials)

Point 8: There were no drop outs? That is unheard of and needs more explanation.

Response 8:  We appreciate the reviewer's attention to participant retention in our study. In the initial version of the manuscript, we inadvertently misreported the number of participants who experienced mortality events. To ensure transparency and provide a more accurate account of the study's progression, we have updated the manuscript as follows:

Originally, a total of 989 participants were recruited at baseline. Regrettably, during our analysis review, we discovered that we had incorrectly reported the mortality events. A total of 33 participants experienced mortality events during the study. Therefore, at the follow-up stage, we had complete and available data for 956 subjects. To clarify, the error was not the omission of exclusions but rather a misreporting of the correct numbers of participants' data analyzed.

To maintain the integrity of our analysis, we applied specific exclusion criteria for different aspects of the study:

  1. For overall and abdominal obesity Receiver Operating Characteristic (ROC) analysis, we excluded 274 participants who were diagnosed as obese and abdominal obese at baseline. Consequently, our assessment was conducted with a reduced sample size of 682 subjects.
  2. In the case of hypertension (HTN) analysis, we excluded 207 subjects who were diagnosed as hypertensive at baseline, resulting in a remaining sample size of 749 individuals for our ROC analysis related to HTN.
  3. Finally, for diabetes mellitus (DM) analysis, we excluded 23 subjects who were diagnosed as diabetic at baseline, leading to a DM analysis conducted with a sample size of 933 subjects.

We understand the importance of clarity in reporting and appreciate the opportunity to provide this clarification. We remain committed to transparency and scientific rigor in our research and reporting.

This has also been incorporated into the manuscript in the results section.

Now it reads: “Out of the initially recruited participants at baseline, 33 participants died, remaining a total of 956 subjects with complete data at the follow-up. Regarding overall and abdominal obesity outcomes, 274 were excluded after applying the exclusion criteria, resulting in a follow-up cohort of 682 participants (figure 1). The inclusion of participants for these outcomes assessment when using a different abdominal obesity criterion can be seen in supplementary Figure A1. For HTN and DM2, 207 and 23 were excluded at baseline, respectively, resulting in follow-up cohorts of 749 and 933 participants for HTN and DM2 assessments (supplementary figures A2 and A3).

Point 9: Table 1: The CI should be values around the mean, but are not? Please explain or revise? Response 9: For clarity issues, Table 1 has been replaced for total and bivariate analysis of compelling data on the original sample of the PERU MIGRANT study.

  1. Obesity: preventing and managing the global epidemic. Report of a WHO consultation. 2000, 894, i–253.
  2. MINSA-INS. GUÍA TÉCNICA PARA LA VALORACIÓN NUTRICIONAL ANTROPOMÉTRICA DE LA PERSONA ADULTA. 2012, 32.
  3. Okorodudu, D.O.; Jumean, M.F.; Montori, V.M.; Romero-Corral, A.; Somers, V.K.; Erwin, P.J.; Lopez-Jimenez, F. Diagnostic performance of body mass index to identify obesity as defined by body adiposity: a systematic review and meta-analysis. International Journal of Obesity 2010, 34, 791-799, doi:10.1038/ijo.2010.5.
  4. Bernabe-Ortiz, A.; Benziger, C.P.; Gilman, R.H.; Smeeth, L.; Miranda, J.J. Sex Differences in Risk Factors for Cardiovascular Disease: The PERU MIGRANT Study. PLOS ONE 2012, 7, e35127, doi:10.1371/journal.pone.0035127.
  5. INEI. Perú: Enfermefades No Transmisibles y Transmisibles, 2022. 2022.
  6. Riley, L.; Guthold, R.; Cowan, M.; Savin, S.; Bhatti, L.; Armstrong, T.; Bonita, R. The World Health Organization STEPwise Approach to Noncommunicable Disease Risk-Factor Surveillance: Methods, Challenges, and Opportunities. American Journal of Public Health 2015, 106, 74-78, doi:10.2105/AJPH.2015.302962.

Reviewer 2 Report

The manuscript is clear, relevant for the field and presented in a well-structured manner.

My specific comments on the manuscript are as follows:

In the introduction part: I have no comments. In the material and methods section: in lines 159-160 it is necessary to supplement protocol number of research approval by the Ethics Committee - this data is missing. In line 157 it is necessary to supplement the limits of the size of skinfolds for the evaluated 4 skinfolds, based on which obesity, the risk of developing hypertension and the development of type 2 diabetes were evaluated, similar to what you mention for waist circumference, BMI ... since the presented study deals with the assessment of the risk of obesity, hypertension and type 2 diabetes using the measurement of skinfolds. There are various conversion indices and equations (e.g., Siri and others) that are used to calculate the amount of subcutaneous fat in relation to obesity. It is necessary to add the methodology by which the measured values ​​of skinfolds were evaluated. Results:   line 172 Figure 1 should be modified graphically; the text is blurry. Discussion: I have no comments. Conclusion: I have no comments. References 52 literary sources were use (with one self-citations); literary source number 53 is missed in the text of manuscript. 21 sources are from the last 5 years; 17 sources for the last 5-10 years and 12 sources that are older than 10 years; 2 sources are listed without year. Based on the above facts, after a minor revision (given specific comments), I recommend publishing the article.

I have no comments

Author Response

Please see the attachment. A revised version of the manuscript was uploaded.

In detail responses to observations are as follows:

Point 1: In the material and methods section: in lines 159-160 it is necessary to supplement protocol number of research approval by the Ethics Committee - this data is missing. In line 157 it is necessary to supplement the limits of the size of skinfolds for the evaluated 4 skinfolds, based on which obesity, the risk of developing hypertension and the development of type 2 diabetes were evaluated, similar to what you mention for waist circumference, BMI ... since the presented study deals with the assessment of the risk of obesity, hypertension and type 2 diabetes using the measurement of skinfolds. There are various conversion indices and equations (e.g., Siri and others) that are used to calculate the amount of subcutaneous fat in relation to obesity. It is necessary to add the methodology by which the measured values of skinfolds were evaluated.

Response 1: The protocol number of research approval by IRB has been added.

Now it reads: “Ethics Committee approval was obtained from the Universidad Peruana Cayetano Heredia before the study commenced (Ethics Committee approval number 60014)”

Regarding skinfold measurements in the present study, we did not convert single skinfold measurements into body fat percentages. Rather we assessed through the receiver operating curve (ROC) analysis the diagnostic capacity for each skinfold measure. Therefore, we did not use the literature skinfold cut-off as was the case for BMI and waist circumference.  Finally, the Youden index was used to determine the best numerical cut-off for each skinfold measure in our study sample.

Nevertheless, for the sake of clarity, we have added this latter information to the statistical analysis section.

Now it reads: “The optimal cut-off point for each skinfold measurement was determined using the Youden index.”

Point 2: line 172 Figure 1 should be modified graphically; the text is blurry

Response 2: We have corrected and updated Figure 1.

Point 3: 52 literary sources were use (with one self-citations); literary source number 53 is missed in the text of manuscript. 21 sources are from the last 5 years; 17 sources for the last 5-10 years and 12 sources that are older than 10 years; 2 sources are listed without year

Response 3: Reference number 53 is now reference number 56 and can be found in line 414. Reference number 30 (IPAQ guidelines) is now reference number 33 and has been corrected.

Reference number 27 (CDC), has been revised and corrected.

Reviewer 3 Report

This is a study that investigated association of skinfold thickness and both abdominal and overall obesity using an existing dataset obtained from a 5-year longitudinal study. Based on the secondary analysis of the dataset that comprised of 715 participants, From the study, the authors proposed sex-specific cut-off points for overall as well as abdominal obesity. In addition, the authors conducted a secondary analysis on risk of developing hypertension and type 2 diabetes mellitus.

While the study has a number of limitations as the authors listed, a consideration to utilize skinfolds as a screening tool for obesity and metabolic abnormality risks is important. Also it was interesting to know that a presence of a 5-year longitudinal study with no drop-out. However, apart from those limitations listed by the authors, there are some additional concerns in the study and it is strongly recommended to clarify following issues and re-analyze the data:

-        Page 3, line 120. It is uncertain how valid BMI and waist circumference cut-off values are for this particular study population. Although the authors explained that they have used “internationally accepted” cut-off values, it does not mean that these cut-off values are valid for this particular population as it is well known that there are ethnic differences in adiposity and BMI or waist circumferences. Therefore, although the study aimed to determine skinfold cut-off values to screen overall and abdominal obesity, there is a strong likelihood that proposed cut-off values may not reflect true adiposity of the study population unless the authors provide scientific evidence that both BMI and waist circumference cut-off values reflect certain levels of adiposity in Peruvian population.

-        Page 3, line 130. While description of anthropometry is a crucial part of the present study, there is a lack of details about its protocol, accuracy and reliability of the data. Please describe more in detail about the protocol of the assessment with a reference. Also, please explain how accurate and precise the results are. Although the authors pointed out that the technical error is unknown, at least the authors can provide how experience the measurer was (i.e. how many years of experience and if the person received a training prior to his/her involvement in the project). In addition, it is inappropriate to use an average value of triplicate measurements as there is a possibility that the average value may be skewed if there is a measurement largely different from other two measurements. It is strongly suggested the authors to re-analyze the results using a median value.

-        P8, line 262. While the authors explained higher sensitivity of sum of skinfolds compared with subscapular skinfolds as the aging effect, considering even distribution of age groups in the present study, this may not be the only reason. It is suggested the authors to consider what are other possible reasons that can contribute observed results.

-        P8, line 272. The authors stated that measuring multiple location skinfolds could minimize the risk of possible abnormal fat distribution in many subjects, why didn’t they aim to determine a single skinfold site with highest AUC in the present study. Rather, the authors should calculate the sum of upper limb skinfolds and the sum of trunk skinfolds so that they can avoid effects of unusual fat accumulation in the screening process. Therefore it is suggested the authors to calculate the sum of upper limb skinfolds and the sum of trunk skinfolds and re-analyze the dataset to propose appropriate sum of skinfolds cut-off points.

-        P9, line 316. This statement appears inaccurate. A number of past studies reported that waist-to-height ratio (WHtR) has been reported to be useful anthropometric index to screen chronic diseases including type 2 diabetes mellitus (T2DM) and metabolic syndrome (MetS) (Suliga et al. 2019; Lee and Yim, 2021). Please consider revise the statement.

Other issues include:

-        Page 2, line 94. Why the authors considered rural-to-urban migrants in the study? How are they different from rural or urban residents? What is the definition of “migrants” (e.g. living in urban for less than a year)?

-        It is better to include prevalence rate of overall and abdominal obesity as measured by BMI and waist circumference in both Tables 1 and 2.

-        P6, line 210. Please check cut-off values for subscapular skinfold and the sum of skinfolds. Aren’t they 18.93 mm and 80.17 mm instead of 17.77 mm and 62.50 mm?

Author Response

Please see the attachment. A revised version of the manuscript was uploaded.

In detail responses to observations are as follows:

Point 1: Page 3, line 120. It is uncertain how valid BMI and waist circumference cut-off values are for this particular study population. Although the authors explained that they have used “internationally accepted” cut-off values, it does not mean that these cut-off values are valid for this particular population as it is well known that there are ethnic differences in adiposity and BMI or waist circumferences. Therefore, although the study aimed to determine skinfold cut-off values to screen overall and abdominal obesity, there is a strong likelihood that proposed cut-off values may not reflect true adiposity of the study population unless the authors provide scientific evidence that both BMI and waist circumference cut-off values reflect certain levels of adiposity in Peruvian population

Response 1: We appreciate the reviewer's consideration of the potential bias about BMI and waist circumference cut-offs used in the study. While we acknowledge that neither BMI nor waist circumference cut-offs for the Peruvian population have been studied for overall and abdominal obesity diagnosis, respectively, it's noteworthy that current Peruvian guidelines for obesity endorse the use of BMI ≥ 30 kg/m² cut-off as the overall obesity criteria index, whereas recommended waist circumference cut-offs (WC ≥102 cm in men and ≥88cm in women) are the same used in the study [1], reinforcing its relevance within our specific study context. Our choice of the selected BMI and waist circumference cut-off as obesity indexes were grounded in its widespread acceptance, alignment with local guidelines, and its ability to effectively identify individuals at a higher risk of obesity-related health complications. However, we also acknowledge the potential limitations, particularly in capturing variations in body composition. In the limitations section as well as in the methods section (variables and measurement) we have added a sentence regarding BMI and waist circumference use as obesity criteria endorsed by current Peruvian guidelines.

In the method section, it reads: “Both definitions have been further endorsed in the technical guidelines for nutritional assessment in Peru.

In the limitation section, it reads: “These criteria are also endorsed by the Peruvian technical guidelines for adult nutritional assessment.

Furthermore, to test the robustness of our analysis, we have included a second abdominal obesity definition based on a Latin American cut-off proposal for detecting visceral adipose tissue ≥ 100 cm2 [2]. The cut-offs were: WC ≥ 94cm in males and ≥ 92 cm in females. This has been stated in the methods section (variables and measurement) which reads: “To test the robustness of diagnostic accuracy results of skinfold thickness for overall and abdominal obesity, additional criterion for defining abdominal obesity was also analyzed (waist circumference ≥ 94cm in males and ≥ 92 cm in females). This criterion was based on a Latin American cut-off proposal for detecting a visceral adipose tissue ≥ 100 cm2

Point 2: Page 3, line 130. While description of anthropometry is a crucial part of the present study, there is a lack of details about its protocol, accuracy and reliability of the data. Please describe more in detail about the protocol of the assessment with a reference. Also, please explain how accurate and precise the results are. Although the authors pointed out that the technical error is unknown, at least the authors can provide how experience the measurer was (i.e. how many years of experience and if the person received a training prior to his/her involvement in the project). In addition, it is inappropriate to use an average value of triplicate measurements as there is a possibility that the average value may be skewed if there is a measurement largely different from other two measurements. It is strongly suggested the authors to re-analyze the results using a median value.

Response 2: A full description has been added to the methods section (variables and measurements) regarding the skinfold assessment protocol.

Now it reads: “The standardized procedure for obtaining each skinfold measurement began with the secure grasp of the skinfold between the thumb and index finger of the observer's left hand, ensuring the inclusion of all underlying adipose tissue within the fold. Subsequently, the skinfold caliper was positioned in the right hand, applied to the skinfold, and maintained at a consistent distance of approximately 1 cm from the fingers of the left hand, ensuring that only the caliper's faces, and not the observer's fingers, applied pressure to the fold. Following the precise placement of the caliper, the observer released the fingers of the right hand, allowing the instrument to exert its maximum pressure while sustaining the grip of the fold with the left hand. Measurements were recorded by reading the dial of the caliper to the nearest 0.2 mm. To enhance accuracy, the caliper was held in place for 2 seconds before each measurement was recorded.

Specific anatomical landmarks guided the placement of each skinfold measurement as follows:

Tricipital: Positioned at the midpoint between the acromion and the olecranon on the posterior surface of the triceps muscle, this measurement was captured with the arm in a relaxed and slightly flexed position, with the palm facing forward, and approximately 1 cm above the designated measurement level.

 Bicipital: Located on the anterior surface of the arm at the same height as the triceps skinfold.

 Subscapular: Attained with the subject in an upright position, their back exposed, and arms at ease by their sides. The fold was taken at a slightly oblique angle, approximately 1 centimeter below the lower angle of the right scapula.

Suprailiac: Situated two centimeters above the left iliac crest, along the mid-axillary line.

 All skinfolds were measured to complete a circuit, and this circuit of measurements was repeated three times. It was ensured that the same skinfold was not measured consecutively, preventing any potential bias or errors in measurement. Finally, the average value from the three measurements of each skinfold was computed and employed for subsequent estimations

 Regarding measurement experience, the standardization process was based on the approach of calculating the average of all observations conducted by the same observer. This was performed monthly before the study began, and each observer field worker was required to measure a minimum of 10 subjects in duplicate. This information has been added to the methods section of the manuscript.

Now it reads: “Regarding the standardization process for anthropometric measurements, this was based on the approach of calculating the average of all observations conducted by the same observer. This was performed monthly before the study began, and each observer field worker was required to measure a minimum of 10 subjects in duplicate.

 Regarding triplicate measurements and the use of the median value, although we agree with the reviewer, individual measurements for each skinfold are lacking in the online freely available dataset of PERU MIGRANT, and only the averaged values are available. Considering this limitation, we have added a sentence in the discussion section (limitations) that accordingly states it.

Now it reads: “The average of triplicate skinfold measures was used instead of the median value as individual values were not available in the free online database, thus preventing us from calculating the median values. As there is a possibility that the average value may be skewed if one of the readings largely differed from the other two, caution is suggested when interpreting the proposed cut-offs. However, anthropometric measurements followed a standardization process based on the approach of calculating the average of all observations conducted by the same observer, potentially minimizing precision errors.

Point 3: P8, line 262. While the authors explained higher sensitivity of sum of skinfolds compared with subscapular skinfolds as the aging effect, considering even distribution of age groups in the present study, this may not be the only reason. It is suggested the authors to consider what are other possible reasons that can contribute observed results.

Response 3: We agree with the reviewer. Considering the new findings after computing different skinfold sums and ratios, as well as re-analyzing the data with an additional abdominal obesity criterion, we have written the paragraph.

Now it reads: “Furthermore, the addition of supra iliac skinfold exhibited a slightly higher specificity than subscapular alone in women. Additionally, when a stricter abdominal obesity definition was used [30], skinfold diagnostic performance remained stable among men, whereas in women, subscapular skinfold and the sum of trunk skinfold increased their AUC and sensitivity. Potential explanations for these findings are related to the aging pattern of body fat redistribution with a characterized reduction in appendicular fat and an increase in trunk fat [38] which may explain why adding the supra iliac to subscapular skinfold slightly increased the AUC both when using the ATP-III and a stricter abdominal obesity criteria. In line with this, a previous study reported that aging increases several times the prevalence of abdominal obesity among men to a greater extent than in women when Peruvian normal-weight individuals were assessed [39]. On the other hand, while aging also contributes to an increase in trunk adiposity accretion among women [40], it is acknowledged that they tend to exhibit higher peripheral to central subcutaneous adiposity [20]. However, it has also been reported that patterns of adipose distribution in women exhibit greater variability among ethnic groups than in men [41]. Furthermore, this may be explained when considering the ethnicity of the study population as a previous study found that among five different Latin American countries, Peruvian and Mexican populations genomics was positively associated with an abdominal fat distribution more severely present in the female sex [21].

Point 4: P8, line 272. The authors stated that measuring multiple location skinfolds could minimize the risk of possible abnormal fat distribution in many subjects, why didn’t they aim to determine a single skinfold site with highest AUC in the present study. Rather, the authors should calculate the sum of upper limb skinfolds and the sum of trunk skinfolds so that they can avoid effects of unusual fat accumulation in the screening process. Therefore it is suggested the authors to calculate the sum of upper limb skinfolds and the sum of trunk skinfolds and re-analyze the dataset to propose appropriate sum of skinfolds cut-off points.

Response 4: We agree with the reviewer. The previous sentence “It is worth considering that while the subscapular skinfold showed better results than the sum of skinfolds overall, measuring multiple location skinfolds could minimize the risk of possible abnormal fat distribution in many subjects” has been deleted.

Now it reads: “While the sum of trunk skinfold and all-four-site skinfold measurements showed similar performance to subscapular skinfold for identifying overall and abdominal obesity in women, it's worth noting that multiple-site skinfold assessment can be time-consuming. Additionally, our study did not observe a significant enhancement in diagnostic accuracy with the inclusion of additional skinfold measurements. Consequently, subscapular skinfold measurement appears to be a clinically useful option for identifying overall and abdominal obesity.

Furthermore, considering the reviewer’s observation, we have computed and analyzed: the sum of the trunk skinfold (subscapular + supra iliac), the sum of the upper limb skinfold (bicipital + tricipital), the upper limb-to-trunk skinfold ratio, and the trunk-to-upper limb skinfold ratio. This has been stated in the methods section.

Now it reads: “The skinfold measurements examined in this study included the bicipital, tricipital, subscapular, supra iliac, the sum of upper limb skinfold (bicipital + tricipital), the sum of trunk skinfold (subscapular + supra iliac), the upper limb-to-trunk skinfold ratio, the trunk-to-upper limb skinfold ratio, and the combined sum of the four site skinfolds.”

All the added tests have been included in the ROC analysis as shown in new tables 2-4, and supplementary table A1.

Point 5: P9, line 316. This statement appears inaccurate. A number of past studies reported that waist-to-height ratio (WHtR) has been reported to be useful anthropometric index to screen chronic diseases including type 2 diabetes mellitus (T2DM) and metabolic syndrome (MetS) (Suliga et al. 2019; Lee and Yim, 2021). Please consider revise the statement.

Response 5: We appreciate the reviewer’s observation. Therefore, in sake of accuracy, we have deleted the sentence.

Point 6: Page 2, line 94. Why the authors considered rural-to-urban migrants in the study? How are they different from rural or urban residents? What is the definition of “migrants” (e.g. living in urban for less than a year)?

Response 6: The decision to include rural-to-urban migrants in our analysis stems from the original study's design, the PERU MIGRANT study, which aimed to evaluate cardiovascular risk based on migration status. This inclusion allows us to maintain consistency with the study's objectives and design.

 As previously reported [3], prior research has shown differences in skinfold measurements between urban dwellers, rural residents, and migrants. However, it's important to clarify that our primary focus in the current study was to assess the diagnostic capacity of skinfold thickness with consideration for differences by sex. Therefore, we did not conduct a ROC (Receiver Operating Characteristic) analysis based on these migration-related differences. Instead, our intention was to provide valuable skinfold thickness cut-off values that could be applied to the broader Peruvian adult population, irrespective of migration status.

The original PERU MIGRANT study defined “migrant” as those participants who were born in the department of Ayacucho (rural area) and were currently living in Lima (urban area) at the time of the study [4].

We have added the latter information in the method section (study design).

Now it reads: “The recruitment of participants permanently living in selected rural and urban areas was done using a single-stage random sampling method. Rural dwellers were selected from San Jose de Secce, Huanta, Ayacucho, while rural-to-urban migrants and urban dwellers were chosen from Las Pampas de San Juan de Miraflores, Lima. The migrant group was considered as those who permanently lived in Las Pampas de San Juan de Miraflores at the time of the baseline assessment but were born in Ayacucho.

Point 7: It is better to include prevalence rate of overall and abdominal obesity as measured by BMI and waist circumference in both Tables 1 and 2.

Response 7: Tables 1 and 2 in the original submission have been deleted. Now Table 1 shows the study population characteristics at the baseline, prior to any exclusion, and stratified by sex. Baseline BMI and waist circumference as continuous variables are shown in Table 1.

Point 8: P6, line 210. Please check cut-off values for subscapular skinfold and the sum of skinfolds. Aren’t they 18.93 mm and 80.17 mm instead of 17.77 mm and 62.50 mm?

Response 8: This has been revised and corrected as the previous sentence that stated, “The optimal cut-off values for subscapular skinfold thickness and the sum of skinfolds were 17.77 mm (sensitivity=68.47%, specificity=71.25%) and 62.50 mm (sensitivity=69.09%, specificity=67.50%), respectively.” has been deleted and replaced considering the new results after adding additional skinfolds sums.

Now it reads “In women, the sum of subscapular + supra iliac skinfold thickness showed the highest AUC of 0.78 (95% CI: 0.72 - 0.83). The optimal cut-off value for subscapular + supra iliac skinfold thickness was 43.00 mm (sensitivity=68.90%, specificity=73.80%). Notably, the subscapular skinfold performed as well as the sum of trunk skinfold, with slightly lower specificity.

  1. MINSA-INS. GUÍA TÉCNICA PARA LA VALORACIÓN NUTRICIONAL ANTROPOMÉTRICA DE LA PERSONA ADULTA. 2012, 32.
  2. Aschner, P.; Buendía, R.; Brajkovich, I.; Gonzalez, A.; Figueredo, R.; Juarez, X.E.; Uriza, F.; Gomez, A.M.; Ponte, C.I. Determination of the cutoff point for waist circumference that establishes the presence of abdominal obesity in Latin American men and women. Diabetes Research and Clinical Practice 2011, 93, 243-247, doi:https://doi.org/10.1016/j.diabres.2011.05.002.
  3. Ruiz-Alejos, A.; Carrillo-Larco, R.M.; Miranda, J.J.; Gilman, R.H.; Smeeth, L.; Bernabé-Ortiz, A. Skinfold thickness and the incidence of type 2 diabetes mellitus and hypertension: an analysis of the PERU MIGRANT study. Public Health Nutrition 2020, 23, 63-71, doi:10.1017/S1368980019001307.
  4. Miranda, J.J.; Gilman, R.H.; García, H.H.; Smeeth, L. The effect on cardiovascular risk factors of migration from rural to urban areas in Peru: PERU MIGRANT Study. BMC Cardiovascular Disorders 2009, 9, 23, doi:10.1186/1471-2261-9-23.

Round 2

Reviewer 1 Report

The authors were responsive to reviewer comments.

Reviewer 3 Report

Thank you very much for considering comments made by all reviewers. The manuscript is now become much clearer. I have no further comments.